# Design and Synthesis of a Mitochondria-Targeting Radioprotectant for Promoting Skin Wound Healing Combined with Ionizing Radiation Injury

**DOI:** 10.3390/ph15060721

**Published:** 2022-06-06

**Authors:** Zaizhi Du, Han Liu, Xie Huang, Yang Li, Liting Wang, Jing Liu, Shuang Long, Rong Li, Qiang Xiang, Shenglin Luo

**Affiliations:** 1State Key Laboratory of Trauma, Burns and Combined Injury, Institute of Combined Injury, Chongqing Engineering Research Center for Nanomedicine, College of Preventive Medicine, Third Military Medical University (Army Medical University), Chongqing 400038, China; duzaizhi@163.com (Z.D.); hx1010033291@hotmail.com (X.H.); slipper_777@163.com (J.L.); ls13648315854@163.com (S.L.); lirongyl@sina.com (R.L.); 2Center of Emergency, First Affiliated Hospital, Third Military Medical University (Army Medical University), Chongqing 400038, China; lhmd2009@163.com (H.L.); lyang5224@163.com (Y.L.); 3Biomedical Analysis Center, Third Military Medical University (Army Medical University), Chongqing 400038, China; onlytea@126.com

**Keywords:** heptamethine cyanine dye, mitochondrion, radiation protection, wound repair

## Abstract

Wound healing is seriously retarded when combined with ionizing radiation injury, because radiation-induced excessive reactive oxygen species (ROS) profoundly affect cell growth and wound healing. Mitochondria play vital roles not only as cellular energy factories but also as the main source of endogenous ROS, and in this work a mitochondria-targeting radioprotectant (CY-TMP1) is reported for radiation injury-combined wound repair. It was designed, synthesized and screened out from different conjugates between mitochondria-targeting heptamethine cyanine dyes and a peroxidation inhibitor 2,2,6,6-tetramethylpiperidinyloxy (TEMPO). CY-TMP1 specifically accumulated in mitochondria, efficiently mitigated mitochondrial ROS and total intracellular ROS induced by 6 Gy of X-ray ionizing irradiation, thereby exhibiting a notable radioprotective effect. The mechanism study further demonstrated that CY-TMP1 protected mitochondria from radiation-induced injury, including maintaining mitochondrial membrane potential (MMP) and ATP generation, thereby reducing the ratio of cell apoptotic death. Particularly, an in vivo experiment showed that CY-TMP1 could effectively accelerate wound closure of mice after 6 Gy of whole-body ionizing radiation. Immunohistochemical staining further indicated that CY-TMP1 may improve wound repair through angiogenesis and re-epithelialization. Therefore, mitochondria-targeting ROS scavengers may present a feasible strategy to conquer refractory wound combined with radiation injury.

## 1. Introduction

Nowadays, millions of patients in the world are distressed with poor wound repair after trauma, burn, surgery or chronic diseases [1]. In particular, refractory wound aggravated by radiation injury has become a common harsh issue since ionizing radiation has been widely used in medical facilities [2,3]. In clinical setting, radiotherapy is one of the most important methods to eliminate cancer cells along with surgery treatment. Despite the significant improvement of intensity-modulated precise radiotherapy, surrounding normal tissues and wound repair are still inevitably influenced by the collateral damage of radiation [4]. There are many factors relevant to the failure of radiation-induced wound healing. These factors are involved in the different repair stages, such as inflammation, proliferation, re-epithelialization and remodeling [5]. Among them, reactive oxygen species (ROS) have been considered as one of the main mechanisms to delay or impede wound healing. The irradiation-induced excessive ROS hinder re-epithelialization and angiogenesis. The abundance of vascular regeneration is an important index to evaluate the quality of skin wound repair, which is closely related to the skin’s nutrition and oxygen supply [6]. Therefore, ameliorating ROS levels is considered to be an important method to promote skin healing of the radiation-combined wound injury.

Ionizing radiation-induced ROS include a class of active free radicals produced from water molecules exposed to ionizing radiation, and secondary free radicals produced from some active intermediates of ROS and cellular components [7]. Although the cell nucleus has long been considered a main target of radiation damage, there is more and more evidence to support the notion that mitochondrion would be another principal subcellular organelle responsible for radiation damage [8]. This is because mitochondrion plays a critical role not only as an energy factory but also as the main source of endogenous ROS [9]. ROS production can be detected in mitochondria a few minutes after cells are exposed to ionizing radiation. The secondary products of mitochondrial ROS, which are closely related to the damage of electron respiratory chains and mitochondrial DNA, were found to be significantly increased [10]. These secondary products are potential factors that cause sustained explosive oxidative stress and the release of a series of apoptotic proteins into the cytoplasm, thereby inducing cell apoptosis and death [11].

Due to the fact that ROS play a vital role in ionizing radiation-induced cell damage and poor wound healing, a lot of natural and synthetic anti-oxidative chemical compounds, hydrogels, biomaterials, nanoparticles with ROS-scavenging ability have been engineered for radiation protection [12,13] or effective repair of skin wound [14,15]. In particular, by mitigating irradiation-induced mitochondrial ROS, different kinds of effective radioprotectants have been discovered [16]. For example, 2,2,6,6-tetramethylpiperidinyloxy (TEMPO, a ROS-scavenging moiety) was chemically conjugated to the mitochondria-targeting bacitracin S [17] or triphenyl phosphorous group [18], radiation-induced apoptosis was proved to significantly decreased. Although these mitochondria-targeting ROS scavengers as potential radioprotectants have been investigated, whether they could also promote wound healing that is combined with refractory radiation injury has not yet been reported.

In this work, we aimed to present a mitochondria-targeting radioprotectant simultaneously for both radioprotection and radiation injury-combined wound repair. It is designed and synthesized by introducing functional TEMPO into mitochondria-targeting heptamethine cyanine dyes [19]. Based on our previous studies on the structure–activity relationship [20], the lipophilic cationic property of the heptamethine core is essential for the mitochondrial targeting. Therefore, different TEMPO-conjugated heptamethine cyanine dyes (CY-TMP) were designed by introducing TEMPO surrounding the heptamethine core. In order to afford an ideal radioprotectant, both the cytotoxicity and radioprotective effect of newly synthesized conjugates were evaluated and compared with Amifostine, a clinically approved radioprotectant drug without mitochondria-targeting ability [21]. Particularly, we investigated whether CY-TMP could effectively accelerate wound closure of mice combined with 6 Gy whole-body ionizing radiation injury. We aim to present a feasible strategy to develop mitochondria-targeting ROS scavengers, which may conquer refractory wounds combined with radiation injury.

## 2. Results and Discussion

### 2.1. Synthesis and Characterization of TEMPO-Conjugated Heptamethine Cyanines

Different TEMPO-conjugated heptamethine cyanine dyes were synthesized by conjugating TEMPO to previously reported mitochondria-targeting heptamethine cyanine dyes [19,22]. As shown in Figure 1a, CY-TMP1 was synthesized by directly substituting the reactive chlorine atom on the heptamethine chain of a heptamethine cyanine dye (compound S3a) by 4-amine-TEMPO. CY-TMP2 and CY-TMP3, conjugated with one or two TEMPO, respectively, were synthesized via carboxamide reaction between carboxyl-containing heptamethine cyanine dyes (Figure 1b,c, compound S4 and compound S3b) and 4-amine-TEMPO. These conjugates were successfully synthesized and confirmed by ^1^H NMR and mass spectrometry (Appendix A). The characteristic EPR spectrum of paramagnetic TEMPO further verified for our successful conjugation (Appendix A). As a typical kind of near-infrared (NIR) dyes, cyanine dyes with a heptamethine core exhibit NIR fluorescent spectra (650–900 nm). According to the absorption and fluorescence spectra, our synthesized TEMPO-conjugated heptamethine cyanine remained to exhibit the characteristic absorption and emission peaks in the NIR region (Appendix A). Thus, TEMPO-conjugated heptamethine cyanines were successfully synthesized.

### 2.2. Evaluation of Radioprotective Effect In Vitro

In order to find an appropriate concentration range for radioprotection, the cytotoxicity of different conjugates was initially tested on human normal liver cells (L-02) using CCK-8 assay. As shown in Figure 1a, the drug concentrations between 0.01–10 μM exhibited no cytotoxicity, except for CY-TMP3. The unexpected result indicates that conjugating more TEMPO moieties could not guarantee a better protective effect. The reason might be that CY-TMP3 may over scavenge physiological ROS which are very indispensable for cell growth and cause cytotoxicity. Among these conjugates, CY-TMP1 did not exhibit any noticeable cytotoxicity even at a high concentration of 25 μM. Amifostine, a well-known FDA-approved radioprotector (a ROS scavenger) in clinic, was chosen to compare cytotoxicity. In contrast to CY-TMP1, Amifostine decreased cell viability by nearly 20% when its concentration was up to 25 μM. Thus, CY-TMP1 with the best biocompatibility was screened out for further radioprotective evaluation. To demonstrate the radioprotective effect, L-02 cells were incubated with different concentrations of CY-TMP1 (0–25 μM) for 6 h and then irradiated for 6 Gy X-ray. The cell viability was detected using CCK-8 assay 72 h after drug incubation. The result showed a 20% decrease in relative cell viability in the irradiation control group compared to the non-irradiation control group. Excitingly, a dose-dependent increase in cell viability was observed for irradiated L-02 cells when treated with CY-TMP1 (1–25 μM). In particular, there was no significant difference in cell viability in the 25 μM CY-TMP1-treated irradiation group compared to the CY-TMP1-treated non-irradiation control group (25 μM CY-TMP1). It is also noteworthy that the radioprotective effect did not occur in the B16-F10 cancer cells. On the contrary, CY-TMP1 not only exhibited a dose-dependent anticancer effect towards B16-F10 cancer cells in the non-irradiation group, but also remarkably sensitized a radiotherapeutic effect (** *p* < 0.01) on B16-F10 cancer cells when exposed to X-ray irradiation (Figure 1b). It is very meaningful that a radioprotector selectively protects normal cells rather than cancer cells from radiation injury during radiotherapy.

Colony-formation assay was further used to confirm the radioprotective effect of CY-TMP1. As shown in Figure 1c, CY-TMP1 or Amifostine alone treatment did not affect cell growth in non-irradiated cells when compared with the control. In contrast, there was nearly no cell survival in the 6 Gy irradiated control group. In the Amifostine group, a bit of cell colony forming was still observed, verifying Amifostine with an appropriate radioprotection effect to alleviate the impairment from radiation. Notably, more cell survival was clearly found in the CY-TMP1 treatment group, indicating more favorable radioprotection of CY-TMP1 than Amifostine.

Excessive ROS generation has been considered as a critical factor in ionizing radiation-induced cell damage, thus intracellular ROS were detected by Reactive Oxygen Species Assay Kit and compared in the different treatment groups. As shown in Figure 2a, ROS levels increased dramatically after irradiation. The increased intracellular ROS were removed efficiently by Amifostine. Impressively, excessive ROS were scavenged in CY-TMP1 group and returned near the normal level. Fluorescent intensity of the ROS probe in different groups was quantified by Image J software, verifying the significantly higher ROS-scavenging ability of CY-TMP1 than Amifostine (*p* < 0.01, Figure 2b).

The potential radioprotection of CY-TMP1 was also investigated by the comet assay and γ-H2A.X immunofluorescent staining, which are typical techniques for evaluation of DNA damage after cells are exposed to ionizing radiation. The tail length of comet as an index of DNA damage was measured with the Comet Assay Software Project (CASP). Both fluorescence imaging and quantitative analysis of comet tail demonstrated that CY-TMP1 effectively alleviated the DNA damage from 6 Gy irradiation (Figure 2c,d). The tail length in the CY-TMP1 group was measured obviously shorter than that in the Amifostine group. Similarly, γ-H2A.X immunofluorescent staining of CY-TMP1-treated L-02 cells displayed significantly lower expression of γ-H2A.X protein (pink spots) than cells in the 6 Gy irradiation control group and the Amifostine-treated group (Figure 2e,f). Based on the above results, CY-TMP1 showed efficient ROS-scavenging ability and radioprotective effect, which was significantly higher than Amifostine.

### 2.3. Mitochondria-Targeting ROS Scavenging and Mechanism Study

The subcellular localization of CY-TMP1 was investigated to uncover its potential mechanism of radioprotective effect. As expected, the synthesized CY-TMP1 was found to specifically accumulate in mitochondria since the red of CY-TMP1 co-localized well with the green of the Mito-Tracker (Figure 3a). As mentioned above, ionizing radiation could induce mitochondrial ROS generation, mitochondria dysfunction and even cell apoptosis. We also examined the mitochondrial ROS-scavenging ability of CY-TMP1 using a Mito-ROS red tracker. Fluorescent imaging of 6 Gy-irradiated cells showed that a large number of mitochondrial ROS were produced. However, the radiation-induced mitochondrial ROS were effectively eliminated in the CY-TMP1-treated group (Figure 3b,c).

Mitochondrial membrane potential (MMP, ΔΨM) is a prerequisite for mitochondrial oxidative phosphorylation to produce ATP and is also one of the key indicators reflecting mitochondrial inner membrane permeability. It has been acknowledged that ionizing radiation could cause the loss of ΔΨM, which leads to a decrease in oxidative metabolism and ATP production. To testify this, MMP was detected with a lipophilic cationic dye (JC-1) and observed by fluorescent microscope. In normal L-02 cells (control group) with high ΔΨM, JC-1 efficiently entered the mitochondria and formed J-aggregates with intense red fluorescence (Figure 3b). However, in the radiated L-02 cells with low ΔΨM, less JC-1 entered the mitochondria and formed monomers with green fluorescence. It is noteworthy that green fluorescence almost disappeared if cells were pretreated with CY-TMP1, indicating that CY-TMP1 protects against the loss of the mitochondrial membrane potential from radiation injury. In addition, ATP generation also decreased significantly after ionizing radiation while CY-TMP1 treatment tended to recover ATP generation from injury (Figure 3d). Therefore, the preliminary mechanism study above indicates that CY-TMP1 may relieve the radiation injury on the mitochondria via targeting and scavenging mitochondrial ROS.

### 2.4. Radiation Protection and Wound Scratch Assay of CY-TMP1 on Human Fibroblast Cells

Before investigation of its effect on skin wound healing in vivo, radiation protection and wound scratch assay of CY-TMP1 were tested on human fibroblast cells (HFF-1). As shown in Figure 4a, CY-TMP1 was demonstrated with a dose-dependent radioprotective effect on HFF-1 cells whereas Amifostine did not have. Calcein AM and PI probes were utilized to sensitively stain live and dead cells, respectively. CY-TMP1 treatment obviously decreased the number of dead HFF-1 cells in the 6 Gy irradiation groups (Figure 4b). According to the results of the wound scratch assay (Figure 4c), the wound closure ability of the 24-h CY-TMP1+6 Gy group was significantly stronger than that of the 6 Gy irradiation group and also stronger than that of the Amifostine+6 Gy group. The fastest wound closure was observed at 48 h in CY-TMP1+6 Gy group.

### 2.5. CY-TMP1 Facilitated Wound Healing Combined with Radiation Injury in Mice

Regarding the radioprotective effect of CY-TMP1 on the mitochondria in vitro, we intended to study its effect on wound healing of mice combined with ionizing radiation injury. Mice were separated into three groups: the control group, 6 Gy radiation group and CY-TMP1 plus 6 Gy radiation group. In the control group, mice were intraperitoneally injected with 100 μL PBS without exposure to radiation. In the CY-TMP1 plus 6 Gy radiation group, 100 μL CY-TMP1 (100 μM in PBS) was subjected to mice 6 h before whole-body ionizing irradiation. Within 30 min after radiation, the full thickness of incisions with a diameter of 1 cm was made on the dorsum of each mouse. As shown in Figure 5a, the control group without radiation showed the fastest wound closure during the observation of two weeks. In contrast, in the 6 Gy radiation group, the wound struggled to heal even on the last day of observation. As expected, much faster wound closure was observed in the CY-TMP1 treatment radiation group than radiation alone group. Quantitative measurement of wound areas determined by Image J software, also verified CY-TMP1 to promote wound healing efficiently (Figure 5b,c). Body weight of all experimental mice was monitored and compared during the treatment (Figure 5d). The result showed a slight decrease in body weight in two radiation groups compared to the control group. On the last day of the two-week observation, there was no significant difference in mouse body weight within all the groups. Collectively, CY-TMP1 could effectively rescue the inhibitory effect of irradiation on wound closure without causing obvious systemic toxicity.

To investigate the inner features of healed wound sites, histological analysis of wound sections was further conducted by hematoxylin and eosin (H&E) staining. As shown in Figure 6a, the images of H&E staining showed a complete wound closure formed in the CY-TMP1-treated wound site, whereas discontinuous epiderm was observed in the 6 Gy radiation-combined wound site. According to Masson and Van Gieson staining, the wounds in the CY-TMP1-treated group were filled with more granulation tissue and collagen than that in the 6 Gy radiation-combined injury group. Subsequently, the expression of platelet endothelial cell adhesion molecule-1 (CD31), a specific maker of newly formed vascular, was detected. Compared to the 6 Gy radiation-combined wound injury group, the expression of CD31 in the CY-TMP1-treated group increased dramatically in the beginning and turned to the normal level as well as the control on the 13th day (Figure 6b). A positive CD31 expression indicates that neovascularization begins to occur. Accordingly, angiogenesis-related cells were sheltered, promoting cell migration and vascular tube formation. Thanks to the ROS-scavenging and angiogenesis-related activities, wound sites could equip themselves with a favorable microenvironment supported by nutrient-providing vessels and peroxidation-regulating assistance, reducing pathological hyperplasia and forming new blood vessels. Thus, our data suggested that CY-TMP1, as a new mitochondria-protective agent, not only promotes wound closure efficiency but also effectively stimulates high-quality regeneration.

## 3. Materials and Methods

### 3.1. Materials

2,3,3-Trimethylindolenine, 1,4-Butanesulfone, 4-Hydroxyphenyl boronic acid and 4-Amine-2,2,6,6-tetramethylpiperidinyloxy (4-amine-TEMPO) were purchased from Adamas; Tetrakis (triphenylphosphine) palladium, N,N-Diisopropylethylamine (DIPEA) and anhydrous ethanol were purchased from Sigma. All reactions were monitored by thin-layer chromatography (TLC). Flash chromatography was carried out using silica gel (200–300 mesh). ^1^H NMR spectra were recorded with an Agilent 600 MHz with trimethylchlorosilane (TMS) as an internal standard. Mass spectrometry (HRMS) was performed on a Bruker Ultraflextreme MALDI-TOF system or Shimadzu MALDI-7090. Electron paramagnetic resonance (EPR) was tested with Bruker A300-10/12. NIR spectroscopy was performed using a NIR spectrophotometer (Shimadzu, UV-3600). Photoluminescence was determined with a NIR fluorescence spectrometer (Lumina Fluorescence Spectrometer, Thermo Fisher, Waltham, MA, USA).

### 3.2. Chemical Synthesis and Structure Characterization

#### 3.2.1. Synthesis of CY-TMP1

Briefly, 2,3,3-trimethylindolenine (3.98 g, 25 mmol) and 1,4-butanesulfone (3.47 g, 25.5 mmol) were added in a 100 mL round-bottom flask and heated to 120 °C for 4 h under nitrogen (N_2_) protection. When the reaction was completed, the reaction solution transformed to solid phase. The solid was dissolved in 10 mL isopropanol and recrystallized with 30 mL acetone to give crude product S2a as a reddish solid (5.29 g, 72% yield). It was used for the next step without further purification. Crude product S2a (500 mg, 1.69 mmol), (*E*)-2-chloro-3-(hydroxymethylene) cyclohex-1-ene-1-carbaldehyde (154 mg, 0.9 mmol) and sodium acetate (147 mg, 1.79 mmol) were dissolved in 10 mL anhydrous ethyl alcohol. The mixture was stirred at 70 °C for 3 h under N_2_ protection. When the reaction was completed, the solution was cooled to ambient temperature, evaporated in vacuum and the residue was purified by column chromatography on silica gel (CH_2_Cl_2_/MeOH = 4:1) to give a green solid S3a (399.5 mg, 65% yield). ^1^H NMR (600 MHz, CD_3_OD) δ 8.44 (d, J = 14.4 Hz, 2H), 7.52 (d, J = 7.2 Hz, 2H), 7.43 (t, J = 7.8 Hz, 2H), 7.38 (d, J = 7.8 Hz, 2H), 7.28 (t, J = 7.2 Hz, 2H), 6.34 (d, J = 14.4 Hz, 2H), 4.23 (t, J = 7.2 Hz, 4H), 2.90 (t, J = 7.2 Hz, 4H), 2.76 (t, J = 6.0 Hz, 4H), 2.04–1.92 (m, 10H), 1.74 (s, 12H); ^13^C NMR (151 MHz, CD_3_OD) δ 172.77, 149.67, 144.17, 142.18, 141.17, 128.50, 126.85, 125.06, 122.05, 110.96, 101.06, 50.35, 49.22, 43.62, 26.92, 25.94, 25.90, 22.18, 20.73.

Compound S3a (110 mg, 0.15 mmol) and 4-amino-TEMPO (75.5 mg, 0.44 mmol) were dissolved in anhydrous DMF (2 mL) and stirred at 75 °C for 4 h under N_2_ protection. When the reaction was completed, the solution was cooled to ambient temperature, evaporated in vacuum and the residue was purified by column chromatography on silica gel (CH_2_Cl_2_/MeOH = 8:1) to give a blue solid CY-TMP1 (31 mg, 24%). ^1^H-NMR δ (600 MHz, CD_3_OD) δ 7.92 (d, J = 10.8 Hz, 2H), 7.37 (d, J = 44.4 Hz, 4H), 7.19–7.14 (m, 4H), 6.05–5.99 (m, 2H), 4.04 (s, 4H), 2.88 (s, 4H), 2.56 (s, 4H), 2.05–1.84 (m, 14H), 1.77–1.57 (m, 13H), 1.42–1.08 (m, 11H), 0.89–0.87 (m, 1H); MALDI-TOF-MS calculated for [M-Na^+^ + 2H^+^] 862.4367, found 862.2347.

#### 3.2.2. Synthesis of CY-TMP2

Compound S3a (117.7 mg, 0.162 mmol), potassium carbonate (47.7 mg, 0.345 mmol) and 4-hydroxyphenylboronic acid (52.1 mg, 0.314 mmol) were dissolved in water (2 mL). The solution was heated to 95 °C and tetrakis (triphenylphosphine) palladium (9.2 mg, 0.008 mmol) was added subsequently. The mixture was continuously stirred for 2 h and monitored by TLC. The reaction solution was cooled to ambient temperature, evaporated in vacuum and the residue was purified by column chromatography on silica gel (CH_2_Cl_2_/MeOH = 4:1) to give a purple solid S4 (120 mg, 91%). ^1^H NMR (600 MHz, CD_3_OD) δ 8.23 (d, J = 7.8 Hz, 2H), 7.35–7.30 (m, 4H), 7.28–7.25 (m, 6H), 7.16 (t, J = 7.2 Hz, 2H), 6.23 (d, J = 13.8 Hz, 2H), 4.12 (t, J = 7.2 Hz, 4H), 2.88 (t, J = 7.2 Hz, 4H), 2.75 (t, J = 6.0 Hz, 4H), 2.06 (p, J = 6.0 Hz, 2H), 1.97–1.89 (m, 8H), 1.17 (s, 12H).

Next, S4 (120 mg, 0.147 mmol), HATU (106.5 mg, 0.28 mmol) and DIPEA (36.2 mg, 0.28 mmol) were dissolved in DMSO (2 mL). The solution was stirred at room temperature for 15 min, then 4-amino-TEMPO (72.35 mg, 0.42 mmol) was added and stirred overnight. The reaction solution was monitored by TLC and purified by column chromatography on silica gel (CH_2_Cl_2_/MeOH = 35:1) to give a green solid CY-TMP2 (76 mg, 54%). ^1^H NMR (600 MHz, DMSO-d6) δ 8.13–8.10 (m, 2H), 7.42–6.91 (m, 12H), 6.22 (s, 2H), 5.70 (s, 1H), 4.07 (s, 4H), 3.33 (s, 6H), 2.65 (s, 4H), 2.44 (s, 12H), 1.90 (s, 2H), 1.69–1.42 (m, 10H), 1.19–0.96 (m, 13H); MALDI-TOF-MS calculated for [M-Na^+^ + H^+^] 965.4, found 965.3.

#### 3.2.3. Synthesis of CY-TMP3

IR-808 (S3b) was prepared based on our previously established method [23]. Then, IR-808 (220 mg, 0.29 mmol), DCC (179 mg, 0.87 mmol) and NHS (100 mg, 0.87 mmol) were dissolved in CH_2_Cl_2_ (5 mL). The solution was stirred at room temperature for 15 min, then 4-amino-TEMPO (108 mg, 0.63 mmol) was added and stirred overnight. The reaction solution was monitored by TLC and purified by column chromatography on silica gel (CH_2_Cl_2_/MeOH = 35:1) to give a green solid CY-TMP3 (126 mg, 39%). MALDI-TOF-MS calculated for [M-Br] 989.6394, found 989.6279.

### 3.3. In Vitro Experimental for Radioprotection Effect Evaluation

#### 3.3.1. Cell Culture and Irradiation

Normal human liver cells (L-02), mouse melanoma cells (B16-F10) and human fibroblast cells (HFF-1) were purchased from the cell bank of the Committee on Type Culture Collection of the Chinese Academy of Sciences (CCTCC). L-02 cells were cultured in RPMI 1640 medium (Gibco, Carlsbad, CA, USA) with 10% fetal bovine serum (FBS, Hyclone, Barrington, IL, USA), 1% penicillin and streptomycin (Gibco). B16-F10 cells and HFF-1 cells were maintained in Dulbecco’s modified Eagle’s medium (DMEM, Gibco) supplemented with 10% fetal bovine serum, 1% penicillin and streptomycin under an atmosphere of 5% CO_2_ at 37 °C. Cells were grown in 6- or 96-well culture plates with 2 × 10^5^ and 2 × 10^3^ cells per well, incubated overnight. Six hours before irradiation, the culture medium was replaced by fresh medium with CY-TMP1 (25 μM) and kept incubated for 24 h. Cells were irradiated with a total dose of 6 Gy at a power of 1.96 Gy/min with an X-ray irradiator (Pxi X-RAD 320, Madison, CT, USA).

#### 3.3.2. CCK-8 Assay

The viability of L-02 cells and B16-F10 cells were evaluated by Cell Counting Kit (CCK-8; MCE, Princeton, NJ, USA) assay. After cells (2 × 10^3^ cells/well) were planted overnight, they were cultured with CY-TMP1, CY-TMP2, CY-TMP3 or Amifostine at the final concentrations of 0, 0.01, 0.1, 1, 10 and 25 μM at 37 °C, respectively. The gradients of CY-TMP1 concentration for incubation of HFF-1 cells were 0, 5 and 10 μM. After being incubated for 6 h, cells in the radiation group were conducted for 6 Gy X-ray ionizing radiation. Cells in 96-well plates were incubated for another 72 h and CCK-8 assay. Cells in each well were incubated with 90 μL complete medium and 10 μL CCK-8, and incubated for 3 h at 37 °C. The optical density (OD) was detected at 450 nm.

#### 3.3.3. Cell Colony-Forming Assay

L-02 cells (1 × 10^3^ cells/well) plated in 6-well plates were cultured overnight. Then, the cells were incubated with Amifostine (5 μM), CY-TMP1 (25 μM) for 6 h. After irradiated with the dose of 6 Gy, cells were allowed to form colonies for one week and washed with PBS before the fixation using 4% paraformaldehyde (Boster, Wuhan, China) for 10 min. Then, the colonies were stained with crystal violet (Beyotime, Shanghai, China), washed with PBS and observed under a light microscope.

#### 3.3.4. Mitochondrial Localization, Mito-ROS and Membrane Potential Assay

To investigate the subcellular localization of CY-TMP1, L-02 cells were incubated with 25 μM CY-TMP1 for 6 h and removed the culture media. Cells were washed three times with PBS and treated cells with Mito-tracker green (Thermo Fisher, Waltham, MA, USA) for 30 min. Cells were promptly washed with PBS three times and fixed with 4% paraformaldehyde for 15 min. Cells were counterstained with DAP, washed with PBS and observed by confocal fluorescent microscope.

Mitochondrial ROS were detected with Mito-ROS Red CMXRos (Beyotime, Shanghai, China). The cells were incubated with CY-TMP1 for 6 h. After irradiation treatment (6 Gy), the culture media was removed and the resting cells were cultured with 1mL Mito-ROS Red CMXRos working solution for 30 min. The cells were washed three times and observed by confocal fluorescent microscope.

Alterations in mitochondrial membrane potential (MMP) were probed with JC-1 (Beyotime) and detected by fluorescence microscope. After irradiation treatment (6 Gy), the culture media was removed and the resting cells were cultured with 1 mL JC-1 working solution at 37 °C for 30 min. Then, the cells were washed three times using PBS and incubated with JC-1 staining buffer (1×). JC-1 staining buffer was removed and the resting cells were cultured with media for fluorescence imaging.

#### 3.3.5. Detection of Reactive Oxygen Species (ROS)

Total ROS were detected with Reactive Oxygen Species Assay Kit (Beyotime). L-02 cells (2 × 10^5^ cells/well) were plated in 6-well plates and cultured overnight. The cells were incubated with CY-YMP1 (10 μM) and Amifostine (10 μM) for 6 h. After irradiated with the 6 Gy for 24 h, cells were promptly washed with PBS and cultured with 10 μM DCFH-DA for 30 min. The intracellular ROS generation was observed by a fluorescence microscope.

#### 3.3.6. Comet Assay

After radiation, L-02 cells (10 μL) were mixed with 75 μL low-melting-point agarose (0.7%) and placed on slides covered with agarose (0.5%). Next, freshly prepared cold lysis buffer was prepared for immersing slides for 1.5 h. Then, sliders were washed with PBS and submerged in a horizontal gel electrophoresis for 1 h. Slides were neutralized three times with cold 0.4 mM Tris-HCl (pH 7.5) for 10 min each time. Finally, slides were cultured with PI, observed by fluorescence microscope and estimated using the Comet Assay Software Project (CASP, Bole, Beijing, China).

#### 3.3.7. Immunofluorescence

After being radiated with 6 Gy for 24 h, cells were fixed with aspirate liquid containing 4% formaldehyde for 15 min under room temperature. Then, cells were washed with PBS, blocked with Blocking Buffer (Beyotime) for 60 min and incubated with phospho-H2A.X (Cell Signaling Technology, Danvers, MA, USA, Phospho-Histone H2A.X (Ser139) Mouse mAb, 80312S, 1:400 dilution) overnight at 4 °C. Cells were promptly washed with PBS and incubated with Anti-Mouse (Cell Signaling, Anti-Mouse IgG (H+L) F(ab’)2 Fragment which was conjugated to Alexa Fluor^®^ 555 fluorescent, 4409, 1:1000 dilution) in the dark for 2 h at room temperature. The nuclei were counterstained with DAPI (MCE,10 μM) in the dark for 10 min at room temperature. Fluorescence was monitored using confocal microscope, and foci of phospho-H2A.X was quantified using ImageJ software (ImageJ v 1.8.0, NIH, Bethesda, MD, USA).

#### 3.3.8. Calcein/PI staining

HFF-1 cells (2 × 10^5^ cells/well) were plated in 6-well plates and cultured overnight. The cells were incubated with CY-YMP1 (10 μM) or Amifostine (10 μM) for 6 h. At 24 h, after being irradiated with the 6 Gy, HFF-1 cells were washed three times with PBS. Then, cells were cultured with Calcein AM and PI (Calcein/PI Cell Viability/Cytotoxicity Assay Kit, Beyotime) at 37 °C for 30 min. The cells were observed under a fluorescence microscope.

#### 3.3.9. Wound Scratch Assay

HFF-1 cells (1 × 10^5^ cells/well) were plated in 24-well plates and cultured overnight. A wound was created by a sterile 200 µL pipet tip. The detached cells were washed away by PBS and cells were incubated with CY-YMP1 (10 μM) or Amifostine (10 μM) for 6 h and irradiated with 6 Gy X-ray. The remaining wound scratch was observed under a light microscope.

### 3.4. In Vivo Experiments for Wound Closure Investigation

#### 3.4.1. Animals

Female BALB/c mice were purchased from Experimental Animal Center of the Third Military Medical University. All of the animals had free access to food and water. All procedures involving animals were reviewed and approved by the Ethics Committee of the Third Military Medical University (Approved No. AMUMEC20210301; Date: 4 March 2021). These mice were separated into three groups: the control group, 6 Gy group and CY-TMP1+6 Gy group. Hair on the dorsum of mice was scraped using electric scissors and removed with a depilatory cream (Veet, ReckittBenckiser, London, UK). One week later, each mouse was intraperitoneally injected with 100 μL PBS or CY-TMP1 (100 μM) and irradiated with 6 Gy ionizing irradiation immediately. Then, the full thickness with a diameter of 1 cm incision was made on the dorsum of each mouse. Photographs were taken on day 0, 2, 4, 6, 8, 10 and 13 after the establishment of an animal model and wound areas were quantitatively measured by Image J software.

#### 3.4.2. Histology and Immunofluorescence Staining

The skin tissues were harvested on the 13th day, and 4% paraformaldehyde was used to fix them overnight. Skin near the wound sites of the mice from each group was entrapped in paraffin and cut into 5 mm-thick sections for H&E, Masson’s trichrome and Van Gieson staining for the investigation. The following immunohistochemical staining of CD31 was performed, and the sections were observed with the same optical microscope. Mice in each group were sacrificed on day 13.

### 3.5. Statistical Analysis

All the experiments were repeated three times and analyzed as mean ± standard deviation (SD). The statistical significance was analyzed by one-way ANOVA analysis using Prism GraphPad 7.0 software. Differences were determined to be significant at * *p* < 0.05, ** *p* < 0.01 and *** *p* < 0.001.

## 4. Conclusions

In summary, we synthesized and identified a mitochondria-targeting agent simultaneously for both radioprotection and radiation injury-combined wound repair. It was found with a notable radioprotective effect, which was better than a clinically approved drug—Amifostine. Particularly, in vivo experiments showed that CY-TMP1 could effectively facilitate wound closure of mice after 6 Gy of whole-body ionizing radiation. Immunohistochemical staining further indicated that CY-TMP1 may improve wound repair through inducing angiogenesis and re-epithelialization. Unrestrained production of ROS and inflammation lead to a hostile healing microenvironment where vascular formation and re-epithelialization will be largely restricted. Therefore, mitochondria-targeting ROS scavengers might represent a feasible strategy to conquer refractory wound combined with radiation injury.

## Data Availability

Data are contained within the article.

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
