# Peer review of "Design and Synthesis of a Mitochondria-Targeting Radioprotectant for Promoting Skin Wound Healing Combined with Ionizing Radiation Injury"

_pharmaceuticals, 2022, doi:10.3390/ph15060721_

Round 1

Reviewer 1 Report

The authors are focused on research of the treatment of skin wounds and designed and synthesized mitochondria-targeted protectant to promote the healing of skin irradiated wounds. The are sufficient amount of methods, however I have several comments and questions:

1/The aims are not written clearly. Results included in aims are not necessary.

2/There are formal mistakes such as N2,Ch2Cl2,CO2, numers of cells and some formal mistakes of references.

3/Some corrections of English are necessary, e.g. in lines 201, 209, 444 and 445.

4/Scheme 2 is not seen clearly.

5/In results it is not necessary to repeat procedures, e.g. lines 283, 284. In lines 294-296 there is a repetition of the procedure. B16-F10 cells were not mentioned in materials.

6/Was CY-TMP1 was not tested individually on skin wounds without irradiation?

7/In methods you mentioned EPH, but results of EPR are missing. 

Reviewer 2 Report

The ms describes the application of  a mitochondria-targeting radioprotectant
(CY-TMP1)  for radiation injury-combined wound repair.

The main question is: why the authors utilized a Normal human liver cells (L-02) and not keratinocytes and fibroblasts? Why the authors did not perform a simply scratch wound assay to test the real wound closure potential? and what about extracellular matrix remodelling and deposition?

More details for statistical analysis are requested.

Round 2

Reviewer 1 Report

The comments of the reviewer were answered therefore I recommend accepting the paper in current form.

Reviewer 2 Report

The authors have correctly replied to questions.